# The Impact of Sustainability Practices on the Going Concern of the Travel and Tourism Industry: Evidence from Developed and Developing Countries

Nagalingam Nagendrakumar *, Kalubowilage Navodya Nilupulee Alwis,
Udage Arachchige Kaveesha Eshani and Seekku Baduge Ushani Kaushalya

Business School, Sri Lanka Institute of Information Technology, New Kandy Rd., Malabe 10115, Sri Lanka
* Correspondence: nagalingam.n@sliit.lk; Tel.: +94-77-228-3199

**Abstract:** Sustainability refers to the evaluation and communication of quantitative and qualitative information of the sustainability performance of a business in a balanced way regarding the environment and the society in which it operates. Companies are responsible for stakeholders' justification and disclosure expenses consisting of dedicated sustainability practices, thereby strengthening the company's financial performance. However, due to the deficiency of consistent information and a lack of transparency in corporate reporting, tourism industries fail to realize the association between sustainability practices and financial performance. Moreover, there is a lack of literature that deals with the impact of macro-level sustainability factors on firms' financial performances. Furthermore, linking the going concern concept and sustainable practices with financial performance through the Z-score model is not frequently done in the corporate world. Hence, this paper investigated the impact of macro-level sustainability practices on the going concern ability in developed and developing countries' tourism industries for the 2016–2020 period, including a sample size of 138 listed companies, through panel data analysis. This study fills the empirical gap by adopting the Altman Z-score to analyze the financial performance related to sustainability practices in terms of environmental, economic, and social dimensions. The empirical results reveal that macro-level sustainability practices significantly impact the going concern from developed and developing countries' perspectives.

**Keywords:** sustainability practices; going concern; financial performance; Altman Z-score model; tourism industry



## 1. Introduction

Sustainability practices are being implemented worldwide and are a key element of the going concern concept in the corporate world. The term sustainability practices was initially defined by Brundtland [1]. Accordingly, its main three dimensions are environmental sustainability, economic sustainability, and social sustainability practices. These dimensions considerably affect businesses and are therefore a mandatory component of corporate strategy.

Thus, sustainability has become a major concern as organizations face an increased level of risk and uncertainty. Currently, most countries tend to prioritize sustainability practices while exploring their potential for better organizational performance. Furthermore, some organizations have adopted sustainability practices as strategic initiatives and measures to enhance their success in terms of financial performance and the reputation of the firm. The main empirical contribution of this study is a knowledge base and a foundation for effective decision making and advice on well-aligned strategies. Hence, this study's main objective was to determine the impact of sustainability practices on the going concern of travel and tourism companies in developed and developing countries.

An organization must find innovative and creative ways to fulfill stakeholders' needs while minimizing the negative effects of operations and products to maintain stability.

However, meeting stakeholders' expectations is challenging, and ensuring sustainability is considerably more complicated than initially thought. Due to the lack of quantitative information, stakeholders of some industries fail to understand the association between social, economic, and environmental sustainability practices and financial performance. Hence, as explained above, organizational stakeholders, including corporate management, fail to understand the significance of sustainability practices in terms of financial performance. Consequently, sustainability practice issues are becoming key components for diverse stakeholders such as corporations, researchers, academics, and policymakers. The above-mentioned empirical study identified a significant impact on sustainability practices on the going concern in businesses. Furthermore, most researchers have proved that sustainability practices significantly impact financial performance. As a result, while contributing to the scarcity of literature, this study differs from previous studies in three keyways.

First, the literature on sustainability practices and going concerns points out several theoretical works, but less research has been conducted on the status of theory concerning the Altman Z-score model. Furthermore, many authors have studied the impact of sustainability practices on financial performance. However, these studies did not adequately consider the Altman Z-score model to evaluate the impact of sustainable practices and financial performance regarding the going concern from the perspective of organizations. Second, this study addresses the macro-level sustainability factors on the going concern. Overall, the present study combines the impact of sustainability practices on the going concern of financial performance concerning the three dimensions using the Altman Z-score model.

The rest of the paper is organized into five distinct stages. Section 2 explains the literature review on the present topic and the background of the study, and then discusses the relevant theories concerning the variables. Section 3 discusses the methodology. Sections 4 and 5 report the data analysis, and discussion respectively. Sections 6 and 7 present the implication, and conclusion respectively.

## 2. Literature Review

Several empirical studies have been conducted in the last decade on the impact of sustainability factors and financial performance. This section presents the progress of literature in the subject area regarding the three key dimensions and helps structure the research path, explaining the independent and dependent variables.

### 2.1. Sustainability Practices on Financial Performance

Many studies have looked at the impact of sustainability practices on the going concern of tourism companies. By adopting panel data analysis, Abdi, et al. [2] investigated the impact of sustainability on the firm value and performance of 27 airlines around the world with data from 2013 to 2019. The findings revealed a positive association between the environmental pillar score and the governance pillar score, with the market-to-book ratio and Tobin's Q being used as proxies for company value and financial performance, respectively. The results suggest that improving both pillars raised the market value and financial efficiency of the airlines under investigation. Rodríguez, et al. [3] investigated the impact of environmental, social, governance, and controversies (ESGC) factors on the financial performance of listed travel and leisure companies by adopting panel data analysis. The results discover a favorable and direct association between sustainability practices and high financial performance in tourist and leisure companies. Additionally, Al-Wattar, et al. [4] observed the relationship between reporting on the dimension of sustainability practices on enhancing the financial performance of hotels on the Iraq Stock Exchange. This study also adopted panel data analysis, and the findings showed that the present accounting system outputs do not entirely fulfill the hotel industry and that there is a decline in reporting information disclosure in annual reports among sustainability practices. Furthermore, these scholars noted that environmental information significantly impacts hotel financial performance more than reporting on economic and social informa-

tion. Finally, their paper validated a positive link between reporting on hotel sustainability and financial performance in the Iraqi hotel sector. Discovering that policymakers should encourage organizations to adopt goals and strategies that are sustainable is a major advantage of the study.

More recently, another study proved the financial success of service sector companies listed on the Bombay Stock Exchange compared with business sustainability performance using panel data analysis [5]. The study findings demonstrate a substantial inverse association between environmental ratings and the selected firms' ROA and return on capital employed (ROCE). Social ratings are the only factor that significantly negatively correlates with return on equity (ROE). Likewise, the composite ratings for the environment, social, and economic governance are adversely significant when compared with ROA and ROCE. The study's practical implications can guide academia, business firms, corporates, policymakers, regulatory authorities, and governments to better understand the relationship and devise policies and strategies accordingly. Additionally, it may encourage businesses to adopt sustainable practices and run more effectively, particularly in rapidly developing nations such as India. Bodhanwala and Bodhanwala [6] examined three tourism-related businesses using an empirical multivariate panel data model to determine how sustainability (ESG) affects firm profitability and market value (transportation, hotel, and leisure). According to the study's findings, among the three tourism-related industries, the hotel sector had the highest ESG compliance, followed by the transportation sector. There appeared to be a sizable positive impact on financial and stock market performance, according to agency and stakeholder theory.

However, several studies have been conducted to determine the impact of sustainability practices on financial performance related to the tourism industry by using various data analysis techniques. Andriyas and Sirait [7] investigated the impact of Cipaku Garden Hotel's sustainability practices and found that the management understood the advantages of the implemented sustainable practices. The results of the interview were then further assessed to determine the state of Cipaku Garden Hotel's sustainable and environmental practices, utilizing initial gap analysis and strength, weakness, opportunity, and threats (SWOT) analysis. This study disclosed that sustainability practices can positively influence a hotel's financial performance and add value to the hotel's organizational environment. It further highlighted that although large firms are often the ones capable of implementing sustainability practices, even small and medium enterprises can apply sustainability practices and benefit from doing so, despite the scarcity of resources. Alzboun, et al. [8] findings showed that sustainability practices do not influence the reduction in Jordanian hotels' financial leakages. However, the results demonstrate that sustainability practices positively affect the hotels' financial leakages by using structural equation modeling and confirmatory factor analysis. Furthermore, Buallay, et al. [9] conducted linear and nonlinear model support, identifying sustainability engagement's effects on tourism industry performance by adopting regression analysis. The empirical findings of the linear models demonstrated that there is a substantial association between ESG and organizational performance. Moreover, the nonlinear model results indicate that the link between sustainability performance and a company's profitability valuation is nonlinear.

Focusing on the previous studies in relation to the impact of sustainability practices on financial performance, apart from the tourism industry, Zyadat [10] disclosed a statistically significant relationship between the sustainability dimension and financial performance, as indicated by return on assets (ROA), earnings per share (EPS), etc. Multiple regression analysis is the most appropriate model to measure the relationship between variables. Moreover, the findings recommended that Islamic banks improve financial performance by implementing appropriate methodologies, and policies were one of the advantages of this research. Similarly, some studies have demonstrated that banks' financial performance has a positive impact on social and environmental sustainability, based on nonlinear threshold regression and cross-sectional linear regression [11]. According to this study, market analysts and investors will gain a better understanding of social sustainability and

how it impacts firm performance in general and bank performance especially. Therefore, financial institutions are encouraged to pursue degrees ranging from altruism to strategic goals by incorporating social sustainability into their business strategic goals and business performance. Moreover, prior research conducted by Pham, et al. [12] observed that sustainability practices have a significantly positive relationship between key performance indicators such as the earnings yield, ROA, and return on equity (ROE) by adopting multiple regression analysis.

*2.2. Environmental Sustainability Practices*

Environmental sustainability practices have become a crucial factor for the corporate world, and most developing and industrialized countries have been denounced for their roles in the destruction of the environment, leading to economic, environmental, and social challenges [13]. Environmental sustainability is described as the responsibility of interacting with the planet to minimize the overall carbon footprints of products by combining multiple competencies. For this purpose, the effects on the four primary natural resources of land, mineral air, water, and energy resources are monitored [14]. Product lifecycle impact assessments can be used to analyze the possible effect on these resources. For example, the company's contribution to regional air quality levels is tracked under air resources. Certification of environmental agreements is used to measure the environmental sustainability of $CO_2$ emissions and electricity production from oil, gas, coal sources, etc. [15].

Environmental sustainability is one of the key dimensions among sustainability practices. Numerous studies have been conducted in relation to environmental sustainability practices on financial performance. Among them, Pulido, et al. [16] found that improving environmental sustainability does not compromise the varying principles of tourism growth, and in fact, the precise opposite is true. The model's findings implied that better policies and regulations will enhance tourism growth, yet this relationship between tourism and environmental sustainability was proven to be bidirectional, because as tourism grows, environmental contamination also rises. The diagonal weight reduction (DWLS) method and the robust standard error and mean and variance correction data analysis method were used for data analysis. Miklosik and Evans [17] investigated how these companies integrate environmental sustainability and protection in their annual report by using 100 sample mining companies listed on the Australian Stock Exchange (ASX). The dimensions of sustainability report disclosure were measured using indicator environmental protection initiatives with Global Reporting Initiative GRI G3 in this index. Descriptive statistics were used to describe the data analysis. To determine if there is a significant relationship with market capitalization, correlation analysis was used. Consequently, the outcome demonstrated that "environmental sustainability management is not the same as other activities; rather, it is powerful, interrelated, and deliberate about environmental security effort is critical". This study introduced and tested a methodology that allows scholars and practitioners to assess the extent to which organizations' environmental initiatives are committed. When compared with other research, this study investigated only 2019, which is a drawback. Moreover, Danso, et al. [18] found an association linking environmental integration and firm financial performance in African (Gahan) small- and medium-sized enterprises. This study used multicollinearity regression analysis. The result reveals an association between financial performance and stakeholder integration, considered by environmental sustainability orientation. The advantage of this research is that environmental sustainability can reduce the consumption of raw materials and waste to develop a firm's reputation to increase revenue.

*2.3. Economic Sustainability Practices*

Economic sustainability practices refer to an organization's effects on its stakeholders' economic situation and local, national, and global economic systems [19]. According to the United Nations Environment Program (UNEP), economic sustainability is defined as the

long-term viability of economic operations, justice, equity distribution benefits, income generation opportunities, and elimination of poverty. In contrast to the above-mentioned definition, most recent studies on sustainability practices have regarded economic sustainability as a unidimensional construct [20]. Rajesh [14] emphasized that companies must maintain their economic viability and health to achieve longevity. Here, they used financial health, potential financial benefits, and trading opportunities as the main criteria to measure a company's economic sustainability.

According to the literature, Qiu, et al. [21] prepared and validated a framework for estimating economic sustainability from local stakeholders' perspectives. They considered a sample size of 2000 citizens in Hong Kong. They gathered data from and conducted in-depth interviews with 12 stakeholders and a telephone survey with 1839 citizens in Hong Kong. The Statistical Package for the Social Science (SPSS) and Analysis of Moment Structure (AMOS) were used to analyze the data. This included constructing tourism economic sustainability consisting of the three dimensions of development control, economic positivity, and personal well-being. This research significantly contributes to the literature on tourism economic sustainability by extending the measurements from the macro-level to the micro-level, thereby revealing the accordance of the resident and expanding the measurements. The evaluation of these dimensions in this paper showed that residents believe that tourism has a significantly positive impact on the community.

### 2.4. Social Sustainability Practices

Social sustainability is a segment of the common trend toward more responsible and ethical business growth, based on social justice, equality, and mutual esteem. Additionally, social sustainability is accomplished through the active support of the formal and informal system, structures, processes, and relationships for the volume of recent and future generations to build a healthy and viable community A more recent approach confirmed that every sustainability feature is social and depends on the interaction between humans and the environment. Under these conditions, social sustainability includes all human activities [22]. Measuring urban social sustainability and its objective contributes to the current literature by constructing an extensive measurement scale for assessing urban social sustainability at the neighborhood level [23]. These researchers argued that urban social sustainability is a multifaceted term with six primary dimensions: Social contact, social participation, sense of place, social engagement, safety, social justice, and neighborhood fulfillment. They gathered data from the household questionnaire titled "Your Neighborhood Living Experience" for a sample of 251 respondents. This paper revealed a sizeable positive link between design quality and overall social responsibility. Furthermore, the findings of this study strengthen the significant role of design improvement strategies in promoting social sustainability and establishing neighborhoods where people now want to live.

### 2.5. Going Concern—Altman Z-Score Model

Going concern is the most important component in the preparation of financial statements and represents a significant turning point in the accounting framework [24]. Many studies have focused on the Altman Z-score model in relation to measuring the going concern of a company. According to the literature, several studies have been conducted by applying the original Altman Z-score model. Nagendrakumar, et al. [25] findings revealed that 91% of hotels are in crisis, 9% are safe, and none are in the gray zone. This study is noteworthy because it provides concrete information on the alarming financial climate, warning hotels in Sri Lanka's tourism industry about the risk of going out of business. Furthermore, few studies have adopted methods other than the Altman Z-score model to measure bankruptcy, such as the Zmijewski analysis (*X*-score), Springate (*S*-score), and Grover analysis (*G*-score). Maharani and Sari [26] findings revealed that five enterprises were in the gray region (gray) category, and 18 companies were in bankruptcy according to Altman analysis (*Z*-score), while four businesses were not in bankruptcy. The collective results demonstrate that the Altman *Z*-score model is capable of foretelling corporate

insolvency. Even though several companies have employed various analytical techniques, some companies remain in the same condition, which is a considerable contribution in this study. Moreover, the tourism industry and its supporting sectors, such as restaurants and hotels, have been significantly impacted by the outbreak of COVID-19. Moreover, hotels, restaurants, and travel companies are some of the sectors most affected by the COVID-19 pandemic. Rahmah and Novianty [27] study analyzed whether there are differences in financial distress before and during the COVID-19 pandemic using the Altman Z-score model. The financial data of 27 companies in 2019 and 2020 that fit the criteria were used. The results showed a negative impact on companies in the hotel, restaurant, and tourism subsectors in Indonesia. Furthermore, the results of this study will probably be used to guide investing decisions. Additionally, this study may be utilized to provide information to businesses who are in financial crisis, enabling them to take urgent action and preserve their operations in the future.

Mahbuba [28] investigated the Z-score model to measure the fundamental financial health of Bangladesh's tannery industry. This study took into account every listed tannery company. Two businesses were in good financial health, according to the research, while the others were not. There are many techniques for measuring the financial health of a business entity. However, Altman's Z-score was proven to be a reliable tool across contexts, as confirmed in this study. The study's findings concluded that it can help managers make financial decisions, investors select investments, and other people protect their interests in the nation's leather business. Another study found that the Altman Z-score model can predict the bankruptcy of companies. This study explained whether the nine samples of face-selected companies were in a gray zone, but not in the bankrupt zone during the 2012–2015 period [29]. This major contribution assists the management of firms in determining the company's financial health and management decision making in avoiding risks related to the company's insolvency. Niresh and Pratheepan [30] examined the likelihood that businesses in Sri Lanka's trading industry may fail, which was the major goal of this study. The study analyzed information from seven trading businesses' most recent five-year financial reports from 2010 to 2014. Companies were categorized into safe, gray, and distressed financial statuses using Altman's initial bankruptcy model. The statistics showed that 71% of trading sector enterprises were experiencing financial difficulties, while the remaining 29% were in the gray zone.

In conclusion, many previous researchers have investigated the impact of sustainability practices on financial performance. However, the impact of macro-level sustainability practices with the Altman Z-score model is yet to be studied. To fill this gap, this research investigated the impact of sustainability practices on the going concern of the tourism industry of developed and developing countries by adopting panel data analysis.

## 3. Methodology

### 3.1. Data Collection and Sample

This section provides a comprehensive discussion of the methodology of the present study. This study relied on secondary data collected from the availability of the information. Sustainability practices data were collected from different platforms such as the World Bank, World Tourism Organization (UNWTO), and Organization for Economic Co-operation and Development (OECD) websites, while going concern data were gathered from the published annual reports of the listed travel and tourism companies on the stock exchanges of the United States of America (USA), Australia, Singapore, Sri Lanka, South Africa, and Malaysia for a period of five years from 2016 to 2020. The gathered data were analyzed through the Stata statistical tool to derive results achieving the study's objectives. The present study used panel data analysis for the analytical process.

This study used several criteria to identify the sample. First, developed and developing countries were selected based on the sustainability ranking issued by the 2021 Sustainable Development Report. Second, this study considered the selected countries' stock-exchange-listed companies that engaged in the travel and tourism industry, since this study wanted

to focus the analysis on travel and tourism companies. Then, 138 listed travel and tourism companies in the selected developed and developing countries were selected as a sample. Table 1 below mentions the selected developed and developing countries, population, and sample of the study.

**Table 1.** Selected countries and list of the population and sample.

| Type | Country | Population | Sample |
|---|---|---|---|
| Developed Countries | United State of America USA | 44 | 32 |
| | Australia | 33 | 24 |
| | Singapore | 23 | 20 |
| Developing Countries | South Africa | 10 | 05 |
| | Malaysia | 30 | 22 |
| | Sri Lanka | 36 | 35 |

For the purpose of this research, this study used a range of variables regarding the company's financial and sustainability performances of the travel and tourism sector. An explanation of these variables is provided in the following sections.

*3.2. Variables*

3.2.1. Dependent Variable

Going concern was the dependent variable in this study, measured by Altman's *Z*-score model. This model has been used in prior studies [25,27,28]. The term going concern is known to businesses that have the resources to continue operation until providing evidence to the contrary. Gheorghe [31] stated that the going concern is the reporting entity, which will continue to operate in the foreseeable future and will be able to recognize assets and discharge financial obligations in the ordinary course of business. Moreover, going concern is a fundamental assumption in financial statement preparation—a company is assumed to have the intention of liquidating or significantly reducing its business scale [32]. Furthermore, it also refers to a company's validity and becomes the underlying assumption in reporting a company's finances, so that if a company has a contrary condition, it becomes problematic. By going concern, the company is expected to be able to defend its activities in the long term rather than being liquidated in the short term.

3.2.2. Altman *Z*-Score Model

In 1968, Edward I Altman developed the *Z*-score model. Since then, it has been used to analyze the financial data from US companies (Huang, et al. [33]) and has been widely recognized as a financial distress model for more than five decades. It is a statistical model that combines five financial ratios to derive a *Z*-score [34], and it predicts a company's financial health. This model has been discovered to be highly efficient for predicting bankruptcy. However, this score has been able to segregate data into two different groups, namely, healthy firms and bankrupt firms [35]. Liquidity, profitability, operating efficiency, and total asset turnover are the key variables of Altman's *Z*-score model.

$$Z = 0.012X1 + 0.014X2 + 0.033X3 + 0.006X4 + 0.999X5 \tag{1}$$

where, X1, X2, X3, X4, and X5 represent liquidity, leverage, productivity, solvency, and efficiency, respectively.

*3.3. Independent Variable*

The independent variable of this study was sustainability practices consisting of the three main dimensions of environmental, economic, and social sustainability practices.

### 3.3.1. Environmental Sustainability

Environmental sustainability is one of the main key dimensions among Sustainability practices, and it includes biological diversity, physical integrity, environmental purity, and resource efficiency [17]. The environmental dimension considers stocks and changes in the stock of environmental assets, often referred to as natural capital, that support tourism or are affected by tourism activities through the provision of environmental resource services. The environmental dimension includes measuring water and energy flow and natural input flow for tourism production processes. Residues generated by the tourism industry and utilization include greenhouse gas (GHG) emissions, solid waste, wastewater, and other pollutants [15].

### 3.3.2. Economic Sustainability

Economic sustainability includes policies that promote long-term economic growth while minimizing the detrimental effects on the environment and culture. Regarding linked products and services, the economic dimension encompasses the production and consumption associated with tourism activities. This is frequently reflected in metrics such as visitor consumption and tourism industry production. The economic dimension also includes a description of the characteristics of the tourism industry and the production processes in the tourism industry [15].

### 3.3.3. Social Sustainability

Social sustainability is a proactive approach to monitoring and recognizing the effects of a company's operations on employees, consumers, value chain workers, and local communities. From a statistical standpoint, accounting for the social dimension in the evaluation of tourism sustainability is the least developed element, but this does not imply that it is unimportant in measurement. There are a number of relatively common themes that measure social sectors (such as health, education, culture, poverty, crime and security, and excellent work), and this is conducted for a population [15].

The below Table 2 presented the summary of the variable's description and operationalization.

**Table 2.** Summary of the variable description and operationalization.

| Concepts | Variables | Dimensions | Indicator | Measurement |
|---|---|---|---|---|
| **Independent Variable** | Sustainability Practices | Environmental Sustainability Practices (ENV) | Modes of Transport (ENV 1) | The sum of no. arrivals by air, water, and land |
| | | | Land Use Area (ENV 2) | Length (in feet) × width (in feet) |
| | | | Air Transport $CO_2$ Emissions (ENV 3) | Fuel consumption × emission factor |
| | | Economic Sustainability Practices (ECON) | Average Daily Spending Per Tourist (ECON 1) | Total annual spending by tourists ÷ total number of annual tourists ÷ 365 |
| | | | Direct Tourism Employment as % of Total Employment (ECON 2) | Total number of residents directly employed by tourism ÷ total size of destination labor force × 100 |
| | | | Travel and Tourism Total Contribution to GDP (ECON 3) | Travel and tourism's total contribution to GDP |
| | | Social Sustainability Practices (SOC) | Income (SOC 1) | The national income of the country ÷ the country's population |
| | | | Health Expenditure for Tourists (SOC 2) | National health expenditure ÷ the country's population |
| | | | Number of Tourists Per 100 Residents (SOC 3) | Total number of tourists × average length of stay ÷ total residents × 365 ÷ 100 |

**Table 2.** *Cont.*

| Concepts | Variables | Dimensions | Indicator | Measurement |
|---|---|---|---|---|
| **Dependent Variable** | Going Concern | Altman Z-score model | X1 (Liquidity) = Working capital/total assets<br>X2 (Leverage) = Retained earnings/total assets<br>X3 (Productivity) = Earnings before Interest & tax/total assets<br>X5 (Solvency) = Market value of Equity/total liabilities<br>X6 (Efficiency) = Sales/total assets | Z = 0.012X1 + 0.014X2 + 0.033X3 + 0.006X4 + 0.999X5 |

Sources: ETIS [36], UNWTO [15], Guerreiro and Seguro [37], and Altman [34].

### 3.4. Data Analysis Techniques

#### 3.4.1. Panel Data Technique

Panel data, known as longitudinal or cross-sectional time series data, constitute one of the most capable statistical tools in econometrics, containing similar variables over certain cross-sections within different periods. Various panel regression approaches and methodologies have been utilized to analyze data to test hypotheses and achieve the given objectives. Descriptive statistics were used to provide a complete description of the impact of sustainability practices on the going concern in developed and developing countries. The FE model and the random effect (RE) model are two models in panel data. The Hausman test must be run to decide which model is appropriate for a study.

#### 3.4.2. Panel Regression with Fixed Effects

This study was based on panel data analysis. In previous studies, this methodology has been adopted [2,3,5,6,38]. FE estimation only considers data on individuals with many observations, and it only estimates the effects for variables that change over time. Panel data are used to solve analytical problems that cannot be answered using only time series or cross-sectional data. In panel data, there are two main methods: FE and RE. FE only analyzes the impact of a variable that changes over time.

#### 3.4.3. Panel Regression with Random Effects

The random effect model's justification is that, unlike the fixed-effect model, it assumes that changes across entitles are random and unrelated to the independent or predictor variables used in the model. Time-invariant variables can serve as explanatory variables under the assumption that the entity's error term is not associated with predictors under random effects.

### 3.5. Diagnostic Test

In a panel data model and a regression model, the diagnostic test identifies various issues. In contrast to traditional OLS regressions, panel regression analysis in Stata does not have a comprehensive set of diagnostic tests, such as the Breusch—Pagan test for panel regression. To run heteroscedasticity testing, additional user-written modules must be downloaded. Different diagnostic tests are designed in econometrics to test multicollinearity, serial correlation, heteroscedasticity, etc. The following sections detail the various tests used in this study.

#### 3.5.1. Variance Inflation Factor Test (VIF)

The variance inflation factor, which assesses the correlation and intensity of correlation between the explanatory variables in a regression model, can be used to detect multicollinearity. Problems with multicollinearity lead to erroneous regression estimates. The higher the VIF, the more collinear the related explanatory variable is with the model's other variables [39].

### 3.5.2. Hausman Test

To decide on which model to be select between the FE and RE, the study carried out the Hausman test, which it is a widely used panel regression techniques in econometrics. According to this test, the null hypothesis is that the preferred model is RM, and the alternative is FE. If the *p*-value is smaller ($p < 0.05$), the null hypothesis is rejected.

### 3.5.3. Breusch–Pagan Test

The Breusch–Pagan test is a heteroscedasticity test. The most common use of heteroscedasticity is in the case of linear regression through the origin. The Breusch–Pagan test specified by the hettest. Heteroscedasticity occurs when the variances of a regression model's error components do not remain constant across the sample data [40] The Breusch–Pagan test has the advantages of being simple to apply and strong when heteroskedasticity is connected to one or more linear proportionality factors [39]. This is used to check the null hypothesis that the error variances are all equal, in opposition to the alternative that the error variances are a multiplicative function.

### 3.6. Empirical Model

In order to pursue the research hypothesis and accomplish the objective of this study, analysis was conducted to examine the impact of environmental, economic, and social sustainability practice indicators on the going concern of travel and tourism companies measured using the Altman Z-score. Following references [3–5], panel data analysis was applied to identify their statistical impact. According to the hypothesis, this study formulated four regression equations in the following empirical models.

The regression equations of the FE model are shown in Table 3:

**Table 3.** Empirical regression models.

| Model | Formula | Explanation |
|---|---|---|
| **Main Model** | $Y_{it} = \beta_0 + \beta_1 X1_{i,t} + \beta_2 X2_{i,t} + \beta_3 X3_{i,t} + \varepsilon_{i,t}$ | i = Company<br>Y = Going concern of the company<br>$\beta_0$ = Constant<br>$\beta_i$ = Coefficient for explanatory variables<br>X1 = Environmental sustainability<br>X2 = Economic sustainability<br>X3 = Social sustainability<br>t = Time period<br>$\varepsilon$ = Error term |
| **Sub Models** | $Y_{it} = \beta_0 + \beta_1 X1.1_{i,t} + \beta_2 X1.2_{i,t} + \beta_3 X1.3_{i,t} + \varepsilon_{i,t}$ | i = Company<br>Y = Going concern of the company<br>$\beta_0$ = Constant<br>$\beta_i$ = Coefficient for explanatory variables<br>X1.1 = Mode of transport<br>X1.2 = Land use area<br>X1.3 = Air transport $CO_2$ emissions<br>t = Time period<br>$\varepsilon$ = Error term |
| | $Y_{it} = \beta_0 + \beta_1 X2.1_{i,t} + \beta_2 X2.2_{i,t} + \beta_3 X2.3_{i,t} + \varepsilon_{i,t}$ | i = Company<br>Y = Going concern of the company<br>$\beta_0$ = Constant<br>$\beta_i$ = Coefficient for explanatory variables<br>X2.1 = Average daily spending per tourist<br>X2.2 = Direct tourism employment as a percentage of total employment<br>X2.3 = Travel and tourism total contribution to gross domestic product (GDP)<br>t = Time period<br>$\varepsilon$ = Error term |
| | $Y_{it} = \beta_0 + \beta_1 X3.1_{i,t} + \beta_2 X3.2_{i,t} + \beta_3 X3.3_{i,t} + \varepsilon_{i,t}$ | i = Company<br>Y = Going concern of the company<br>$\beta_0$ = Constant<br>$\beta_i$ = Coefficient for explanatory variables<br>X3.1 = Income<br>X3.2 = Health expenditure for tourists<br>X3.3 = The number of tourists per 100 residents<br>t = Time period<br>$\varepsilon$ = Error term |

## 4. Results

### 4.1. Multicollinearity Diagnostic Test

In Table 4, the multicollinearity test explains the inter-correlation between the variables. If VIF has a value of 1, it means that the independent variable is not correlated with other variables. If VIF has a value between 1 and 5, it means it is moderately correlated. A VIF value greater than 5 means that the independent variable is highly correlated with other variables. As this research proved the above-stated conditions, it affirms that there is a high correlation between income, air transport, $CO_2$ emissions, and health expenditure per tourist. The other indicators of sustainability practices did not have the multicollinearity issue in this model and, hence, specify that the regression results are more reliable.

**Table 4.** Multicollinearity diagnostic test—VIF values.

| Variables | VIF |
|---|---|
| **Sustainability Practices** | |
| Ln Modes of Transport | 3.78 |
| Land Use Area | 3.10 |
| Ln Air Transport $CO_2$ Emissions | 7.31 |
| **Mean of Environmental Sustainability Practices VIF** | **4.73** |
| Ln Average Daily Spending Per Tourist | 1.30 |
| Direct Tourism Employment as % of Total Employment | 1.13 |
| Ln Travel and Tourism Total Contribution to GDP | 1.15 |
| **Mean of Economic Sustainability Practices VIF** | **1.20** |
| Ln Income | 8.40 |
| Ln Health Expenditure for Tourists | 8.83 |
| Number of Tourists Per 100 Residents | 1.20 |
| **Mean of Social Sustainability Practices VIF** | **6.14** |

Note: VIF = variation inflation factor test; GDP = gross domestic product.

### 4.2. Heteroskedasticity Test (Breusch–Pagan Test)

The study's Breusch—Pagan/Cook—Weisberg test yielded a value of 0.0000, which means that the *p*-value was significant and rejected the null hypothesis. Finally, it accepted the alternative hypothesis, indicating heteroskedasticity in the dataset. The results of the heteroskedasticity test are presented in Table 5.

**Table 5.** Heteroskedasticity test results.

| | Z-Score |
|---|---|
| | **Coef.** |
| Chi-square test value | 52.64 |
| *p*-Value | 0.0000 |

Note: Coef. = coefficient.

### 4.3. Hausman Test

Table 6 illustrates the Hausman results of the study. The Hausman test had to be run to decide the appropriate model, i.e., FE or RE, for the study. According to this study, the null hypothesis was RE, and the alternative hypothesis was FE. The *p*-value was less than 0.05, which indicates significance. Therefore, this study's null hypothesis of using RE was rejected. Finally, the results reveal that the alternative hypothesis of using FE was suitable for this study.

**Table 6.** Hausman test results.

| | Z-Score |
|---|---|
| | Coef. |
| Chi-square test value | 51.76 |
| *p*-value | 0.0000 |

Note: Coef. = Coefficient.

*4.4. Descriptive Statistics*

Table 7 shows a summary of the descriptive analysis for the dependent and independent variables in developed and developing countries for the 2016–2020 period used in this study. The descriptive statistics present a better understanding of the objectives. The *Z*-score was the dependent variable of this study, and the mean and standard deviation were 0.4235 and 0.2551, respectively. The mean and standard deviation of the mode of transport were 31,900,000 and 43,800,000, respectively. However, the mode of transport represents the collection of the three categories of land, water, and air, thus indicating how many people traveled by land, water, and air. The land use area had a mean of 3,705,943 km$^2$, and the standard deviation was 4,309,540 km$^2$. The mean and standard deviation of air transport $CO_2$ emissions were 16.130 and 1.6894, respectively. Each indicator represented a mean and standard deviation of 9748.22, 13,111.71, 0.0640, 0.0692, 75,900 million (Mn), and 120,000 Mn, respectively. The mean income in Sri Lankan Rupees (LKR) was 5.9 Mn and the standard deviation was 4.7 Mn. Furthermore, the mean health expenditure for tourists (annual) was LKR 5 trillion, while the standard deviation was LKR 6.27 Tn. The mean and standard deviation of the number of tourists per 100 residents were 49.047 and 48.6237, respectively.

**Table 7.** Summary of the descriptive statistics.

| Variable | Going Concern | Environmental Sustainability Practices | | | Economic Sustainability Practices | | | Social Sustainability Practices | | |
|---|---|---|---|---|---|---|---|---|---|---|
| | Z-Score | ENV 1 ('000) | ENV 2 | ENV 3 | ECON 1 | ECON 2 | ECON 3 | SOC 1 | SOC 2 | SOC 3 |
| Obs. | 690 | 690 | 690 | 690 | 690 | 690 | 690 | 690 | 690 | 690 |
| Mean | 0.4235 | 31,900,000 | 3,705,943 | 37,800,000 | 9748.2 | 0.0640 | 75,900 | 5986.54 | 5,410,000 | 49.047 |
| Std. Dev. | 0.2551 | 43,800,000 | 4,309,540 | 62,200,000 | 13,111.7 | 0.0692 | 120,000 | 4753.33 | 6,270,000 | 48.6237 |
| Min | 0.3752 | 508,000 | 709 | 521,944 | 1357.09 | 0.0166 | 1830 | 402.90 | 5680 | 3.285 |
| Max | 1.0855 | 170,000,000 | 9,831,510 | 207,000,000 | 65,660.1 | 0.2257 | 319,000 | 12,000 | 18,900,000 | 171.1231 |

Note: '000 = Thousands; Obs. = observation; Std. Dev. = standard deviation; Min. = minimum; Max. = maximum; ENV 1= mode of transport; ENV 2 = land use area; ENV 3 = air transport CO2 emissions; ECON 1 = average daily spending per tourist; ECON 2 = direct tourism employment as a % of total employment,; ECON 3 = travel and tourism's total contribution to GDP (Mn); SOC 1 = income; SOC 2 = health expenditure per tourist; SOC 3 = number of tourists per 100 residents.

*4.5. Regression Analysis*

The regression analysis findings for the impact of environmental sustainability practices on the going concern of the travel and tourism listed companies are demonstrated in Table 8. The results show that the environmental sustainability indicators, i.e., ENV 1, ENV 2, and ENV 3, have a significant and positive impact on the companies' *Z*-score values. This implies that an increase in all three indicators leads to a higher *Z*-score value. Therefore, the tourism industry's investment in environmental sustainability practices, such as reducing emissions, using re-usable resources, having proper garbage disposal methods, implementing environment policy, and maximizing shareholder benefits, may also result in a higher *Z*-score value. As a result, companies are also expected to maintain their profitability. Considering the numeric data according to the results, the *R*-squared value of FE was 0.8693, emphasizing that 86.93% of the total variance of the going concern

of the travel and tourism companies can be explained by environmental sustainability practices. The coefficients of ENV 1, ENV 2, and ENV 3 were 0.161306, 0.0000084, and 0.08457, respectively. The significant coefficients of the mode of transport, land use area, and air transport $CO_2$ emissions were significant at the 1% significance level. The conclusion is that environmental sustainability practices strongly impact the going concern of the listed travel and tourism companies. Hence, the null hypothesis of this study was rejected, since the $p$-value is less than 0.05. Therefore, the alternative hypothesis was accepted, and the following regression equation was constructed:

$$\text{Yit} = -8.327125 + 0.161306 \text{ X1.1i, t} + 0.00000084 \text{ X1.2i, t} + 0.08457 \text{ X1.3i, t} + \varepsilon\text{i, t} \quad (2)$$

**Table 8.** Impact of environment sustainability practices on going concern of the travel and tourism companies.

| Dependent Variable | | | Ln Z-Score |
|---|---|---|---|
| **Independent Variables** | **Dimensions** | **Indicators** | **Countries** |
| | | | **FE** |
| **Sustainability Practices** | Environmental Sustainability Practices | Ln ENV 1 | 0.161306 (0.0220276) *** |
| | | ENV 2 | 0.000000840 ($12 \times 10^{-7}$) *** |
| | | Ln ENV 3 | 0.08457 (0.15864) *** |
| **Constant** | | | −8.327125 (0.609188) *** |
| **R²** | | | 0.8693 |
| **Prob > F** | | | 0.0000 |

*** $p < 0.01$. Note: This study analyzed the data by FE, which is a fixed-effect model. The figures without parentheses are the coefficients, while those within parentheses are the standard errors. ENV 1 = mode of transport; ENV 2 = land use area; ENV 3 = air transport CO2 emissions.

Table 9 depicts the regression results of the economic sustainability practices and the going concern of the travel and tourism companies, where the *R*-squared value of 0.8728 indicates an 87.28% variation in the going concern. This is justified by economic sustainability practices. When considering the individual impact of economic sustainability, the going concern of the travel and tourism companies decreased by 0.24428 and 0.44581 when ECON 1 and ECON 3 were changed. Additionally, the going concern of the travel and tourism companies increased by 13.52663 when ECON 2 was changed. However, the significant impact of these three economic sustainability indicators demonstrates that ECON 1 was significant at the 1% significance level, while the other two indicators were significant at the 5% significance level. The results imply that economic sustainability practices have a strong significant impact on the going concern of the travel and tourism companies. According to the results, ECON 1 and ECON 3 had a negative and significant impact on the companies' Z-score value. At the same time, ECON 2 had a positive and significant impact on the companies' Z-score value. Therefore, the changes in these three indicators show that the Z-score value either decreased or increased, indicating that investing in ECON 1 and ECON 3 leads to disadvantages and financial distress in travel and tourism companies. In contrast, investing in ECON 2 leads to an advantage and increases profitability in the tourism industry. This study rejected the null hypothesis and accepted the alternative hypothesis, because the *p*-value was less than 0.05. The following regression equation was developed based on these results:

$$\text{Yit} = 10.48374 - 0.24428\text{X2.1i, t} + 13.52663 \text{ X2.2i, t} - 0.44581 \text{ X2.3i, t} + \varepsilon\text{i, t} \quad (3)$$

**Table 9.** Impact of economic sustainability practices on the going concern of the travel and tourism companies.

| Dependent Variable | | | Ln Z-Score |
|---|---|---|---|
| **Independent Variables** | **Dimensions** | **Indicators** | **Countries** |
| | | | **FE** |
| **Sustainability Practices** | Economic Sustainability Practices | Ln ECON 1 | −0.24428 (0.02532) *** |
| | | ECON 2 | 13.52663 (6.33171) ** |
| | | Ln ECON 3 | −0.44581 (0.19581) ** |
| **Constant** | | | 10.48374 (4.32159) *** |
| **R²** | | | 0.8728 |
| **Prob > F** | | | 0.0000 |

** $p < 0.05$, and *** $p < 0.01$. Note: This study analyzed the data by FE, which is a fixed-effect model. The figures without parentheses are the coefficients, while those within parentheses are the standard errors. ECON 1 = average daily spending per tourist; ECON 2 = direct tourism employment as a % of total employment; ECON 3 = travel and tourism's total contribution to GDP (Mn).

Table 10 shows the results of the regression analysis for the travel and tourism companies' continued going concern for social sustainable practices. The results demonstrate that social sustainability practices were responsible for an 88.44% variance in the going concern, with an *R*-squared value of 0.8844. The related coefficients SOC 1, SOC 2, and SOC 3 were 0.59942, 0.47443, and −0.00276, respectively. The social sustainable indicators were significant at the 1% significance level. The findings show that social sustainability practices had a strong significant impact on the listed travel and tourism companies. Considering the social sustainability results, SOC 1 and SOC 3 had a negative and significant impact on the companies' *Z*-score value. Meanwhile, SOC 2 had a positive and significant impact on the companies' *Z*-score value. Therefore, the changes in these three indicators show that the Z-score value either decreased or increased, denoting that investing in SOC1 and SOC 3 will bring about a disadvantage and financial distress. Furthermore, investing in SOC 2 will result in potential profit-making opportunities for the tourism industry. Social sustainability suggests that investors appreciate companies who are perceived to better handle their social responsibilities. The *p*-value for this study was less than 0.05; therefore, the null hypothesis was rejected, and the alternative hypothesis was accepted. Thus, this study constructed the following equation:

$$Y_{it} = -5.16568 - 0.59942\,X3.1_{i,t} + 0.47443\,X3.2_{i,t} - 0.00276\,X3.3_{i,t} + \varepsilon_{i,t} \qquad (4)$$

Table 11 shows the regression results of the main model, which indicates the sustainability practices on the going concern of the travel and tourism companies that are listed on the stock exchanges of the USA, Australia, Singapore, South Africa, Sri Lanka, and Malaysia. The *R*-squared value in FE demonstrates that 88.06% of the variance in the going concern on the travel and tourism companies was explained by sustainability practices. The constant of this model was identified with a coefficient of 8.46775, which became significant at the 1% level. According to this study, the going concern of the travel and tourism companies was mainly explained by the above-stated three sustainability practice dimensions over five years. Based on the model results, the impact of environmental sustainability practices on the going concern on the travel and tourism companies were significant at the 1% significance level. This indicates that environmental sustainability practices had a significant positive impact on the going concern. The impact of economic and social sustainability practices on the going concern of the travel and tourism companies

was significant at the 1% level, which indicates that economic and social sustainability practices had a significant negative impact on the going concern of the companies. The higher Z-score value suggests that the travel and tourism companies will exist in the long term. This is because investors apply the Z-score to decide whether to buy or sell company stocks, depending on the financial health of a company. Therefore, this study can conclude with the following regression equation according to the results:

$$Y_{it} = 8.46775 + 0.000000541 \, X1_{i,\,t} - 0.35163 \, X2_{i,\,t} - 0.00425 \, X3_{i,\,t} + \varepsilon_{i,\,t} \tag{5}$$

**Table 10.** Impact of social sustainability practices on the going concern of the travel and tourism companies.

| Dependent Variable | | | Ln Z-Score |
|---|---|---|---|
| **Independent Variables** | **Dimensions** | **Indicators** | **Countries** |
| | | | **FE** |
| **Sustainability Practices** | Social Sustainability Practices | Ln SOC 1 | −0.59942 (0.08572) *** |
| | | Ln SOC 2 | 0.47443 (0.04159) *** |
| | | SOC 3 | −0.00276 (0.00092) *** |
| **Constant** | | | −5.16568 (1.72012) *** |
| **R²** | | | 0.8844 |
| **Prob > F** | | | 0.0000 |

*** $p < 0.01$. Note: This study analyzed the data by FE, which is a fixed-effect model. The figures without parentheses are the coefficients, while those within parentheses are the standard errors. SOC 1 = income; SOC 2 = health expenditure per tourist; SOC 3 = number of tourists per 100 residents.

**Table 11.** Impact of sustainability practices on the going concern of the travel and tourism companies.

| Dependent Variable | | Ln Z-Score |
|---|---|---|
| **Independent Variables** | **Dimensions** | **Countries** |
| | | **FE** |
| **Sustainability Practices** | Environmental Sustainability Practices | 0.000000541 (0.000000115) *** |
| | Economic Sustainability Practices | −0.35163 (0.03608) *** |
| | Social Sustainability Practices | −0.00425 (0.00112) *** |
| **Constant** | | 8.46775 (1.42702) *** |
| **R²** | | 0.8806 |
| **Prob > F** | | 0.0000 |

*** $p < 0.01$. Note: This study analyzed the data by FE, which is a fixed-effect model. The figures without parentheses are the coefficients, while those within parentheses are the standard errors.

Table 12 depicts the present status of all the hypotheses proposed for the study.

**Table 12.** Status of the hypotheses developed for the study.

| Hypotheses | Statues |
|---|---|
| **H1ₐ** | Accepted |
| **H2ₐ** | Accepted |
| **H3ₐ** | Accepted |

Source: authors' compilation.

## 5. Discussion

This research was conducted to identify the impact of sustainability practices on the going concern of the travel and tourism companies listed on the stock exchanges in developed and developing countries. The empirical findings demonstrated that sustainability practices have had a strong significant impact on the going concern of travel and tourism companies. These findings are aligned with the previous literature [2,3], which differs in terms of how the dependent variable was measured, whereas the present study has adopted the Z-score as the measure. However, previous researchers Pham, et al. [12] and Jyoti and Khanna [5] studied sustainability practices extensively with traditional financial performance methods such as ROA, ROE, and EPS. Laskar, et al. [41] found that corporate sustainability practices significantly and positively impact financial performance by using the market-to-book ratio (MBR) and Tobin's Q in the Indian and Japanese contexts. Bodhanwala and Bodhanwala [38] indicated that there is a significantly positive relationship between the sustainability and financial performance of firms. In addition, the findings demonstrated that companies using prominent sustainable development strategies record higher profitability and have a significantly lower gearing level. This highlights the strategic benefits of adopting sustainability practices that improve a business's survival and stability beyond just financial performance.

To the best of the researchers' knowledge, no past research has been conducted combining macro-level sustainability practices and going concern. Thus, filling this gap, the present research examined macro-economic sustainability factors and firms going concern in the travel and tourism industry in developed and developing countries using the Z-score as the analysis tool, capturing these in a single study. Three hypotheses developed for the current study confirmed that there is a significant impact of environmental, economic, and social sustainability practices on the going concern.

**H1$_a$.** *There is a significant impact of environmental sustainability on the going concern of the listed tourism companies in developed and developing countries.*

The current study results show that there is a positive impact of environmental sustainability practices on the going concern. Tan, et al. [42] results reveal that environmental performance has a positive impact on the financial performance of the hotel industry but not on other firms related to the tourism industry. The sample size was relatively small; therefore, these scholars recommended that future researchers expand the sample size while using publicly trading tourism-related firms across diverse economic regions. Molina-Azorín, et al. [43] indicated that environmental sustainability has a significant impact on firms' performance in the Spanish hotel industry. On the contrary, Manrique and Martí-Ballester [44] found that environmental practices have a significant and positive impact on corporate financial performance in developed and developing countries. This means that good environmental management helps to increase sales and improve financial performance alongside reputation. Dzomonda and Fatoki [45] found that the reduction in carbon emissions was significantly positively related to earnings per share price. The above-mentioned studies adopted various indicators to measure environmental sustainability practices when compared with the present study's environmental indicators. The positive impact of the environmental sustainability dimension on the going concern indicates that environmentally friendly activities add value to businesses, eventually materializing as improved financial performance.

**H2$_a$.** *There is a significant impact of economic sustainability on the going concern of the listed tourism companies in developed and developing countries.*

When considering economic sustainability practices, the results affirm that they have a strong and significant impact on the going concern of travel and tourism companies. However, past studies that have argued the impact of economic sustainability practices on firms' performance are limited, although such studies have revealed that economic sustainability practices have a positive effect on a firm's financial performance. According

to past researchers, Kasbun, et al. [46] suggested that economic sustainability positively affects financial performance, measured using ROA and ROE. Recent researchers Rahi, et al. [47] found a positive effect on governance practices and financial performance while using ROE, return on shares, return on invested capital (ROIC), and EPS. Additionally, the findings revealed that firms face a risk in adopting sustainability practices, as they follow a logic that contradicts the purely economic rationale. Al-Dhaimesh and Kamel Al Zobi [48] validated this result with a statistically significant effect of economic sustainability practices on financial performance. Additionally, they stated that economic sustainability dimensions improve a firm's financial performance by improving the confidence of potential investors and creditors.

**H3$_a$.** *There is a significant impact of social sustainability on the going concern of the listed tourism companies in developed and developing countries.*

The evidence presented in this paper shows that social sustainability practices have a moderate impact on the going concern of travel and tourism companies. Dimitrova [22] developed a more recent approach that confirms that every feature of sustainability is truly social and depends on the interaction between humans and the environment. Rodríguez and del Mar Armas Cruz [49] carried out research focusing on the Spanish hotel sector. Their findings indicate a strong and positive relationship between social sustainability and the Spanish hotel sector. According to Schönborn, et al. [50], this proves that there is a positive relationship between corporate social sustainability culture and a company's financial success. Furthermore, Erhinyoja and Marcella [51] showed a negative relationship between financial performance and corporate social sustainability in the Nigerian oil and gas industry. Hence, sustainable reporting practices should be encouraged among Nigerian companies, aligning existing global sustainability standards to reflect social and environmental challenges. When compared with previous studies, this paper went in a different direction for stakeholders, while assessing income and health expenditure per tourist. However, social sustainability supports the financial performance of companies. By proactively managing social sustainability, managers and stakeholders are assisted to achieve their objectives, which can easily support maintenance of the integrity of their businesses.

## 6. Implications

### 6.1. Theoretical Implications

This paper is an initial attempt to add to the limited existing knowledge about the Altman Z-score, taking into consideration the concept of going concern, which is rarely tested in terms of sustainability practices. In other words, the existing theoretical framework is based on a limited quantity of literature. This study proposed a conceptualization framework to explore the impact of sustainability practices on the going concern. The findings are expected to assist tourism companies in exploring the continuity of their company's long-term future without bankruptcy and maintaining profit stability. Furthermore, this research will help to minimize adverse economic, social, and environmental impacts on financial performance.

### 6.2. Managerial Implications

This study's findings highlight practical implications for tourism industry companies in developed and developing countries, encouraging them to adopt sustainability practices directly or indirectly to enhance their financial performance. It is critical that the managers and owners of tourism companies adopt and implement good sustainability policies and performance if the former wants to build, retain, and maintain their market position in a highly competitive market. Tourism is considered an industry volatile to the external environment (economic shocks, booms, world tourism trends, etc.) and a customer-oriented industry that evolves continuously. Businesses that adopt sustainability practices can improve their organizational resilience and survive external shocks, especially in times of uncertainty, much better than other firms. Therefore, this study supports the stakeholders

of tourism companies to enrich their know-how about financial investments (feasibility and returns) and improve their well-being and sustainability. Adopting such practices enables a company's management to make decisions efficiently and effectively and, most importantly, to enhance business value as a good corporate citizen in the long run.

## 7. Conclusions

In conclusion, the absence of sustainability practices in a company's financial records implies a need for better awareness and knowledge of the subject. The findings of this study may persuade the tourism industry to add environmental, economic, and social sustainability practices, since they improve the financial performance of companies. Another realization is that decisionmakers must understand the advantages of enhancing sustainability practices for the tourism industry. Policymakers would have a more comprehensive view of the elements affecting shareholders' wealth maximization principles in the tourism industry.

Descriptive statistics and regression analysis were adopted to achieve the objectives of the present study. The regression results of the present study conclude that there is a strong significant impact of sustainability practices on the going concern of developed and developing countries' travel and tourism industries. Sustainability practices have a strong and significant impact on the going concern of travel and tourism companies, environmental sustainability practices have a positive significant impact on the going concern of travel and tourism companies, and economic and social sustainability practices have a strong and significant impact on the going concern of travel and tourism companies.

The present study's findings have implications for researchers from practical and theoretical perspectives. From an academic point of view, the impact of adopting sustainability practices on a company's financial performance remains debatable and limited, despite development in the tourism literature. An issue of the current study is the lack of knowledge of stakeholders related to sustainability practices and financial performance: What is the impact of sustainability practices on the going concern of listed companies in developed and developing countries? Nevertheless, the sample of this study and time frame could be able to answer this question and add to the body of literature. Hence, future researchers can conduct this study on one selected industry or as a combination of these industries. Second, there are more indicators related to and influencing sustainability dimensions, but this study only included a few indicators, because information is not available in the annual reports of some companies. Furthermore, the scope of this study can be expanded in the future by adding new sustainable practices; thus, the findings can be generalized. Third, this study was based only on five years (2016–2020). Researchers could conduct new research by increasing the duration of the present study to gain a broader picture. Another limitation is that the total population of this study consisted of six developed and developing countries conducted using secondary data. Based on these studies, future researchers may have the possibility to fill a potential research gap by conducting their investigations using primary data and adjusting the sample population, industry sectors, and regions.

The key contribution of this study is increasing the financial performance of companies while focusing on sustainability practices. Thus, managers must be competent and have a positive outlook in order to successfully lead financial performance and sustainability practices and generate financial benefits for organizations. Consequently, it is the manager's responsibility to develop a plan that considers sustainability practices. In other words, managers should work according to their objectives. Second, the business must strike a balance between its financial performance and sustainability practices. Furthermore, customer-oriented industries evolve continuously.

**Author Contributions:** N.N. Supervision, writing—review and editing, conceptualization, and project administration; K.N.N.A. Conceptualization, data curation, formal analysis, investigation, methodology, resources, software, validation, visualization, and writing—original draft; U.A.K.E. Conceptualization, data curation, formal analysis, investigation, methodology, resources, software,

validation, visualization, and writing—original draft; S.B.U.K. Conceptualization, data curation, formal analysis, investigation, methodology, resources, software, validation, visualization, and writing—original draft. All authors have read and agreed to the published version of the manuscript.

**Funding:** This research received no external funding.

**Institutional Review Board Statement:** Not applicable.

**Informed Consent Statement:** Not applicable.

**Data Availability Statement:** DOI: https://doi.org/10.5281/zenodo.7031676.

**Conflicts of Interest:** The authors declare no conflict of interest.

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
