# Peer review of "The Impact of Sustainability Practices on the Going Concern of the Travel and Tourism Industry: Evidence from Developed and Developing Countries"

_sustainability, doi:10.3390/su142417046_

Round 1

Reviewer 1 Report

I find the topic of the paper pretty interesting, and the methodology and results are explained well. However, two main features need to be improved:

1. Theoretical explanation. More recent studies are needed in order to explain the topic.

2. Conclusions. The two paragraph used to write down the conclusions are too general (it is quite similar to the abstract).  

Author Response

Dear respected reviewer,

Thank you very much for your valuable and insightful comments on our paper. We took them as very seriously and have given the maximum possible correction and have uploaded them for your concern, please.

Reviewer 2 Report

This work reflects the research method adequately, however the suggestion is to improve the references related to the dependent variable, including more Five references, not earlier than 2015.

Please include 5 more references related to sustainability practices in terms of environment, economic and social dimensions. for Turism or business related with tourism.

Author Response

(The authors gave the same response as above.)

Reviewer 3 Report

The authors propose a study on the impact of sustainability practices on the going concern of travel and tourism industry. At a current state the paper is very confusing both in relation to the introductory part but also on the chapters relating to the description of the data, methodology and results. The paper therefore requires a great deal of effort to be ready for publication. However, considering the topic of interest and the quality of the contents of the final part of the paper, my decision is to accept the paper with major revision. Here are my suggestions for improving the paper.

MAJOR COMMENTS

-       - The literature review section is poor of contents. More studies should be added with a more consistent reference to the aim of the study and the methodology proposed by the Authors. Furthermore, the literature review aims to justify the motivation of the study and, eventually, the research gap that is to be filled. However, Chapter 2 is missing all of these aspects.

-    -    All the references in the paper, following the recommendation of the Journal, should be numbered in order of appearance, and indicated by a numeral in square brackets (e.g. [1], [3-6], [6,7]).

-     -   Il line 141 clarify what “FE” stands for.

-     -   Equations should be numbered.

-  -      I suggest summarizing the final considerations of section 3.1 in a table in order to make easier the explanation of each formula.

-   -     The Methodology section is also lacking in the description of the method presented. Furthermore, studies that analyse similar issues with the same method should be cited with the aim of reinforcing research choices. I suggest adding more considerations about that.

- -       Line 171, “past literature”, but which literature? It is necessary to mention it.

- -       There are two “Table 1”. Moreover, even if the “first” table is quite useful in the understanding of the meaning of the variables, it should be introduced and described in the relative section.

- -       I don't understand the contents of section 5. Multicollinearity Diagnostic Test (and the others) should be explained and explained in the methodology section with an appropriate sub-section devoted to the description of the tests used. Also what are VIF, FE and RE? what do the acronyms stand for? Please, be more consistent.

-   -     At the beginning of section 5.4 I think the reference to the next table is missing.

--        Improve the readability of table 5.

-   -     The section devoted to results is very confusing and it is difficult to understand its contents easily. From what I can understand, the results of the statistical analysis do not seem very good (especially for R squared). Authors should endeavour to better organize this section and provide a better explanation of the results obtained.

-     -   Overall, I suggest an English check for the whole paper.

MINOR COMMENTS

- In section 4.1 and 4.2 remove the dot before the title.

- Line 166 remove the dot after “main dimensions”.

- Remove the dot at the beginning of line 243.

- In line 324 remove “Error! Reference source not found

Author Response

(The authors gave the same response as above.)

Reviewer 4 Report

The manuscript analyses the financial performance related to sustainability practices in terms of environment, economic and social dimensions by adopting the Altman Z score method. The research is meaningful for tourism industry sustainable operations. The data sources are reliable and the method used in the manuscript is suitable to answer the research questions. The results revealed that macro-level sustainability  practices  significantly  impact  the  going  concern,  from  the  developed  and  developing countries' perspectives. For improving the quality of the manuscript, some major revisions need to be done as follows.

1)      The abstract need to be rehearsed with research objectives, data, methodologies and results and contributions to the tourism study.

2)      The data are obtained from the listed 138 travel and  tourism companies.  The data scale need to be explained.  How to adjust the data to the same scale of the 690 observations?  How to match the environmental data to other data?

3)      The result part needs to be written clearly.

4)      The written style and grammar need to be checked carefully.

Author Response

(The authors gave the same response as above.)

Round 2

Reviewer 3 Report

The Authors addressed all the issues that I hilighted. In my opinion, at the current state, the paper is ready for publication. 

Reviewer 4 Report

The manuscript has been revised regarding the sugesstions and it suitable to be published in our journal after revised the Abstract in which the background need to be decreased and the results and conclusion need to be emphasised.